# Roots and Shoots: A Pilot Parallel Randomised Controlled Trial Assessing the Feasibility and Acceptability of a Nature-Based Self-Help Intervention for Low Wellbeing

**DOI:** 10.3390/bs15081096

**Published:** 2025-08-12

**Authors:** Matthew Owens, Chloe Houghton, Paige Beattie, Hannah L. I. Bunce

**Affiliations:** 1Department of Psychology, The Mood Disorders Centre, University of Exeter, Exeter EX4 4QQ, UK; chloehoughton15@icloud.com (C.H.); paige.beattie8@gmail.com (P.B.); 2The ROWAN Group, Exeter, UK; hannah.bunce@somersetft.nhs.uk; 3Neurology, Department of Neurosciences, Somerset Foundation Trust, Taunton TA1 5DA, UK

**Keywords:** nature, wellbeing, pilot randomised controlled trial

## Abstract

The burden of depression is a public health concern, and traditional treatment approaches to mental health alone may be insufficient. The effects of contact with nature on wellbeing have been shown to reduce stress and improve mood, emotional wellbeing and mental health difficulties. Thus, self-guided nature-based interventions (NBIs) present a promising approach to improving mental health and wellbeing. However, there is limited evidence on the development of such interventions. This two-armed pilot randomised controlled trial aimed at determining the feasibility, acceptability and preliminary efficacy of a novel, 4-week, self-help NBI (*Roots and Shoots*©). Forty-seven participants were randomised (1:1) to either receive the *Roots and Shoots* intervention or a waitlist control. Participants in both conditions completed measures including wellbeing (primary outcome) and depressive symptoms, rumination, sleep and nature relatedness (secondary outcomes) at three timepoints: baseline (T0), 2 weeks (T1) and 4 weeks (T2). Those who completed the intervention period reported high acceptability and satisfaction with the intervention. The findings from this pilot study indicate potential for improvements in wellbeing following the intervention, which appears reasonably feasible and acceptable. Future research is warranted to further investigate the efficacy of this novel NBI in a larger, powered clinical trial.

## 1. Introduction

Mental health difficulties, including depression, are highly prevalent and on the rise ([60]). They are associated with high levels of disability globally ([44]; [99]) and have significant social impacts ([81]). In addition, associated economic costs are large, approximately GBP 118 billion per year in the UK ([56]). Traditionally, mental healthcare has been driven by our collective “rule of rescue”, focusing primarily on treatment ([11]) and almost exclusively on symptom reduction ([30]). Unfortunately, however, the majority of individuals in need of treatment receive none ([86]).

Although the barriers to treatment are complex ([93]), a long-standing issue remains that UK health services are under significant strain ([59]). This has led to alternative approaches being explored, including efforts at preventing mental health difficulties ([25]) and focusing on lifestyle factors ([32]; [75]), which include access to natural environments ([23]; [67]). At the same time, there has been a trend to complement traditional pathogenic health service models with salutogenic approaches. These aim to strengthen resources that enable individuals to promote their health and wellbeing when presented with stressors or adversity ([36]). Low wellbeing is linked to mental health difficulties, as evidenced by analysis of the UK Office of National Statistics (ONS) data showing that people who report low subjective wellbeing are twice as likely to report major depression ([47]). Furthermore, one cohort study found that those with low levels of positive wellbeing at the start of the study had an increased likelihood of developing depression ten years later ([98]). This suggests that low wellbeing may foreshadow the unfolding of mental ill health and may therefore be an important target of intervention to mitigate future difficulties. Thus, there have been growing research efforts towards testing positive psychology interventions that not only seek to alleviate depressive symptoms but also promote the enhancement of wellbeing ([14]; [22]; [90]). This shift away from a treatment-only model may decrease perceived barriers to accessing mental health support, such as stigma, accessibility and the preference to be self-reliant ([35]).

As a society, we have less connection to the natural world, with 50% of the global population living in urban areas ([17]), and we spend up to 90% of our time indoors ([34]). An associated concern here is that the loss of relational connection to natural environments may lead to poor wellbeing over the generations ([12]). Circumstantial evidence suggests an association between urban/natural environments and mental health. For example, urban living is associated with a higher reported number of individuals receiving prescriptions for psychotropic medications for mental health disorders ([58]) and is associated with a higher burden of depression ([91]). Conversely, rural countries have lower incident rates of mental health disorders than urbanised countries ([91]).

The extant literature on natural environments suggests that increasing contact and connection to natural environments is beneficial for human health and wellbeing ([8]; [37]; [101]), and prospective studies on nature exposure suggest that access to greenspace in childhood predicts a lower risk of mental health disorder in adulthood ([27]). Furthermore, nature-based interventions (NBI; [24]), including nature-based therapies ([84]), have shown promise in improving mental health and wellbeing. For example, in a recent meta-analysis, it was suggested that individuals who engaged with forest therapy interventions had a 17-fold increased likelihood of reaching remission and were three times more likely to have a 50% or greater reduction in depressive symptoms than those who continued with usual care ([73]).

NBIs can range from full contact with wild natural environments, such as forest bathing (the Japanese art of Shinrin Yoku; [2]), to incorporating clinical psychological approaches, such as CBT ([84]), mindfulness ([24]) or blended approaches ([57]). We report here the results of a pilot parallel-groups randomised controlled trial (RCT) testing the feasibility and acceptability of a nature-based self-help intervention versus waitlist control to improve mental wellbeing in adults.

## 2. Materials and Methods

### 2.1. Study Design and Sample Size

This study was a two-armed parallel pilot randomised controlled trial (RCT) that aimed to determine the feasibility and acceptability of a novel self-help guidebook, *Roots and Shoots*, designed to deliver NBI for improving wellbeing, compared against a waitlist control. To be included in the study, participants needed to be resident in the UK, report a mental wellbeing score of 22 or less on the Short Warwick–Edinburgh Mental Wellbeing Scale (SWEMWBS) at screening and have at least average self-reported English language ability. Exclusion criteria included having any pharmacological or psychological treatment for mental health in the last three months. Participants completed the study remotely, and the intervention guidebook (a physical A5-sized booklet) was posted to the participants’ home address. The intervention was designed as a four-week hybrid programme of both real-world and online activities to be completed remotely. Participants were asked to complete measures at baseline (T0), 2 weeks (T1) and 4 weeks (T2, study end). Ethical approval was granted by a university psychology ethics committee prior to collection of data (Application ID: 5412277), and the study was preregistered on clinicaltrials.gov (Identifier: NCT06513793).

Formal power calculations are not recommended for pilot studies ([97]), given that one of their functions is to gain initial estimates for sample size in future trials ([48]). We used recommended rules of thumb ([97]) to determine sample size. We reasoned that small to medium effect sizes in favour of the intervention would be worth detecting and made an a priori decision to set a target sample size of n = 40. We assumed a 10% attrition rate over the course of the study and therefore aimed to randomise ≥ 44 participants into the trial.

### 2.2. Procedure

Between May and July of 2024, 786 potential participants responded to recruitment advertisements and completed an eligibility screen using the online survey platform Qualtrics^**®**^. In an attempt to reduce bot completion of the screener, we used a CAPTCHA (Completely Automated Public Turing Test to Tell Computers and Humans Apart) item at the start of the online screener ([100]). Of those participants screened, 264 (33.59%) were excluded because they did not meet the inclusion criteria, leaving a pool of 522 (66.41%) potentially eligible applicants. On review of this pool of participants, one declined to participate further, several IP addresses or GPS coordinates were found to be originating from outside the UK (n = 13), and there were a number of suspected bots identified (n = 345). Finally, a number of participants were excluded for not providing their postal address (n = 67).

Eligible participants were contacted via email by a member of the research team to notify them of their eligibility status and confirm their availability to participate in the study. A total of 47 participants were randomly assigned to the trial arms (1:1) using random permuted blocks: intervention (n = 24) or waitlist control (n = 23). Randomisation was conducted using the online randomisation service sealedenvelope.com. This is an automated process, which means researchers were blind to the allocation process, preventing selection bias. Informed consent was requested initially, followed by completion of baseline measures (T0) after receipt of the guidebook (sent by post). Participants assigned to the intervention group were directed to adhere to the weekly activities and engage with additional online resources housed on a companion website. All participants were asked to complete measures at a 2-week interval (T1) and immediately post-intervention at week 4 (T2). Both the intervention and waitlist control groups completed the same set of online questionnaires at all three timepoints (T0, T1 and T2) with the exception of five additional feasibility and acceptability questions included in the final survey (T2) for participants in the intervention condition.

In an attempt to reduce participant attrition, up to three follow-up reminders were sent via email to participants who did not initially respond to the questionnaire or provided incomplete responses. After completion of T2 assessments, all participants received debrief information, which included the study aims, researcher contact details and signposting for further support and mental health services. Participants received a GBP 10 Amazon gift voucher in remuneration upon completion of the study. Primary medical practitioner details were taken from participants for management of any reports of self-harm or suicidal ideation. Clinical cover to manage risk was provided by a Health and Care Professions Council (HCPC)-registered clinical psychologist.

Initially, five participants did not respond to their assessment of baseline measures. Therefore, 42 participants, all of whom received the intended allocation, provided at least partial data for the analysis. One participant formally withdrew, and 11 were lost to follow-up. Therefore, 30 participants completed the study (13 in the waitlist arm and 17 in the intervention arm), which gives an overall attrition rate of 36%; 43% (13/23) and 29% (17/24) for the waitlist and intervention arms, respectively.

The timeline for treatment and assessments is provided in the CONSORT diagram in Figure 1.

### 2.3. Trial Arms

#### 2.3.1. Nature-Based Intervention: Roots and Shoots

The intervention, *Roots and Shoots*, consisted of a printed, A5-sized self-help guidebook covering six main sections and weekly, evidence-based activities. Designed as a short, four-week course for improving low wellbeing, the guidebook incorporates principles of nature-based psychology with techniques and exercises informed by several clinical psychological therapies. In the first section, participants were provided with a comprehensive introduction to the course, including psychoeducation around nature and wellbeing. Psychoeducation is important to ensure participants understand the information necessary to engage with the intervention, including relevant underlying theory. This is consistent with research that has highlighted the benefits of psychoeducation as an effective therapeutic approach for improving low mood in depressed individuals ([89]). The following sections guided participants through a set of weekly activities, including a habit wheel (tracking identified nature-based habits to foster), a coping tool (identifying alternative coping behaviours), a honeycomb challenge (tracking contact with nature), mood and nature connection monitoring and an expressive writing exercise. The rationale for the weekly exercises was informed by behavioural activation principles ([54]), mindfulness ([38]), cultivation of habits ([26]), self-tracking of mood ([92]), nature exposure ([9]), compassion-focused therapy ([65]) and acceptance and commitment therapy (ACT) techniques ([83]). Throughout *Roots and Shoots*, participants were signposted to an online webspace, where they had access to additional digital resources, including nature videos, meditations and breathing exercises. These activities draw on evidence showing the beneficial effects of virtual nature exposure ([10]; [62]), nature-guided imagery ([45]; [64]), brief nature-based meditation ([13]) and guided breathwork ([31]). Finally, participants were encouraged to reflect on their time spent engaging with nature and the guidebook activities.

#### 2.3.2. Waitlist Control

Participants in the waitlist control group completed the assessments at the three time points: baseline, at a 2-week interval and at 4 weeks post baseline, after which they received a copy of the *Roots and Shoots* guidebook and were provided full access to the additional online resources but were not monitored.

### 2.4. Acceptability and Feasibility

We assessed acceptability in terms of the perceived importance, relevance and helpfulness of the intervention (e.g., “Overall I enjoyed doing the *Roots and Shoots* Programme”), as well as motivation and willingness to recommend *Roots and Shoots* to peers. In addition, the specific elements of the intervention were assessed (e.g., “The habit wheel was helpful”). Feasibility was similarly assessed with self-report items (e.g., “The guidebook activities were easy to do”). Participants were asked to rate these bespoke items using a 5-point categorical scale: *Strongly disagree* (1) *Disagree* (2), *Neutral* (3), *Agree* (4) and *Strongly agree* (5). Finally, participants were invited to give free-text responses for any comments and suggestions about the intervention: “We really value your opinions and feedback on *Roots and Shoots*. Please let us know any additional comments about your experience with the intervention (for example, what would you like to see more of, or less of, or anything you think we should add) that you would like to share with us.” We also report the ability to recruit to target, which assessed the demand for the NBI. Finally, we provide estimates on the preliminary efficacy ([6]) on the primary outcome, as well as clinically significant change (CSC).

### 2.5. Outcome Measures

As made clear in the trial register, the primary outcome measure was the Short Warwick–Edinburgh Mental Wellbeing Scale (SWEMWBS).

#### 2.5.1. The Short Warwick–Edinburgh Mental Wellbeing Scale (SWEMWBS)

This is a unidimensional scale measuring mental wellbeing that captures social, eudemonic and hedonic dimensions of wellbeing ([85]). As a widely used scale, it has been well validated in general ([85]) and in secondary care mental health user samples ([4]). The short version (SWEMWBS) includes seven positively phrased items, such as “I’ve been feeling optimistic about the future”, that are rated on a five-point Likert scale ranging from 1 (“None of the time”) to 5 (“All of the time”). In an extensive meta-analysis by Mack and colleagues ([52]), the SWEMWBS has been shown to have good internal consistency, with an average Cronbach’s alpha of 0.86. A score of ≤19.5 is indicative of possible mild depression ([78]), and a score of ≥27.5 suggests high wellbeing.

The following secondary outcomes were included.

#### 2.5.2. Generalised Anxiety Disorder (GAD7)

The Generalised Anxiety Disorder 7-item scale (GAD7) is a brief, self-administered screening tool designed to measure anxiety severity based on DSM-IV criteria ([82]). This is a clinical measure, and a score between 7 and 10 is indicative of generalised anxiety disorder ([68]).

#### 2.5.3. Patient Health Questionnaire (PHQ8)

The PHQ8 is a well-validated clinical measure of symptoms of depression. A score of ≥10 on PHQ8 represents probable caseness for clinical depression ([46]).

#### 2.5.4. Perceived Stress Scale (PSS4)

The Perceived Stress Scale (PSS4) is a widely used self-reported measure assessing perceived stress levels over the past month. The scale is a shortened version of the original, and it has shown reliable validity in measuring perceived stress in diverse populations ([95]).

#### 2.5.5. Ruminative Response Scale (RSS)

The RSS aims to measure ruminative tendency in response to low mood. We used the brooding subscale, consisting of five items ([88]).

#### 2.5.6. Jenkins Sleep Scale (JSS)

The JSS ([41]) is a brief 4-item measure of subjective sleep quality that aims to assess common sleep difficulties over the past month. Recent meta-analyses suggest the JSS demonstrates good reliability, reporting a mean Cronbach’s alpha of 0.80 ([40]).

#### 2.5.7. Five Facet Mindfulness Questionnaire (FFMQ-15)

The Five Facet Mindfulness Questionnaire 15 (FFMQ-15) is a short-form self-report instrument that measures mindfulness through five dimensions: observing, describing, acting with awareness, nonjudging and nonreactivity. This 15-item version provides a quick yet comprehensive assessment of mindfulness. It is recommended to omit the observing items when deriving a total score ([3]).

#### 2.5.8. Nature Relatedness Scale (NR-6)

The short-form version of the Nature Relatedness Scale is a six-item scale measuring the individual’s sense of connection with the natural environment ([61]).

### 2.6. Statistical Analysis

We report the means, standard deviations and 95% confidence intervals for the primary and secondary outcomes and report Hedges’ g, given the small sample size, for estimates of effect size on the mean differences.$$Hedges’\;g=Cohen’s\;d\times \left(1-\frac{3}{4\left(n_{1}+n_{2}-2\right)-1}\right)$$
where $n_{1}+n_{2}$ = sample size for each arm. An online calculator was used to calculate Hedges’ g from the pooled estimates taken from the multiple imputation datasets that were generated via the mice package for R.

We tested the preliminary effectiveness of the intervention on the primary outcome using the *lmer* function and adding participant ID as a random factor. We first compared a main effects model with an interaction model (Trial arm X Time) and then followed up with linear models within each arm. We also calculated reliable change (RC) and clinically significant change (CSC) to help estimate the clinical significance of any improvement seen in the primary outcome from T0 to T2. The former (RC) asks whether an individual’s change in score is greater than that expected due to measurement error, and the latter (CSC) asks whether the magnitude of change is likely to be clinically meaningful. Here, the level of functioning after the intervention should be outside the range (two standard deviations) of a clinical population in the direction of functionality ([39]). For RC calculations, we used the Cronbach’s alpha of 0.86 reported by Mack and colleagues ([52]). For CSC (criterion a), we used the clinical and normative data from the SWEMWBS (https://warwick.ac.uk/fac/sci/med/research/platform/wemwbs/using/howto/ (accessed on 13 May 2025)): clinical (mean = 18.66, standard deviation = 2.50) and normative (mean = 23.72, standard deviation = 3.20).

We followed the principle of intention to treat (ITT) in the analysis, which gives unbiased estimates of the efficacy of an intervention ([55]); that is, we did not exclude any participants based on their level of engagement with the intervention. To perform the ITT, we used multiple imputation using the mice package for R to account for missing data (100 imputed datasets with maximum iterations = 20). Attrition was determined by calculating the proportions of participants retained throughout the study.

## 3. Results

### 3.1. Characteristics of the Participants

There were no significant differences between trial arms on the demographic variables (see Table 1).

### 3.2. Acceptability and Feasibility

The guidebook was generally well received by participants in the intervention arm. All respondents (100%) reported that they enjoyed using the guidebook, and 88% either agreed or strongly agreed that the intervention was helpful in improving their wellbeing. The majority indicated they would use the guidebook again in the future (94%) and would recommend it to a friend (94%). Furthermore, 100% of participants found the guidebook activities easy to use and follow, and 82% of participants found the online activities easy to use, with 88% finding them easy to follow. In addition, most participants reported finding each of the individual components of the interventions (psychoeducation, habit wheel, coping tool, honeycomb challenge, mood and nature connection tracking, expressive writing, coping tool, honeycomb challenge, nature videos, meditations and breathing exercises) useful. Self-reported adherence to the guidebook was good, with the majority of participants (88%) reporting completing the guidebook for at least 50% of the days and more than half (65%) for over 75% of the days. Overall, 47% of participants indicated that they agreed or strongly agreed that four weeks was an adequate length for the intervention, 47% reported it should be longer, and 6% reported it should be shorter. Full results of the acceptability and feasibility items can be found in Table 2. Qualitative feedback was largely positive, for example, “*I really enjoyed the program, I actually found out I am really good at growing things! And I’ve been able to get out more into nature and my garden and just enjoy it and relax a lot more. I think it’s great little program and should be definitely considered by doctors to forward [sic] for mental help*”.

### 3.3. Quantitative Outcomes

The imputed estimates for the mean scores on the primary and secondary outcomes are shown in Table 3. The differences between the trial arms on baseline measures (T0) were small, suggesting that the randomisation procedure was successful. The baseline mean for mental wellbeing was approximately 19 points for both arms, suggesting that participants on average had low wellbeing at the point of commencing the study. Wellbeing had increased by an estimated four points at T1 and approximately five points at T2. The effect size comparing the relative change between arms was large at T1 (*g* = 1.02), gaining in size slightly at T2 (*g* = 1.24). There was a general pattern of increasing effect sizes from T1 to T2 in the secondary outcomes, with the exception of stress and sleep. In terms of the other clinical measures used in this study (PHQ8 and GAD7), average scores for the arms suggested that the sample was characterised by clinical depression and generalised anxiety disorder at T0. The effect size at T2 was medium for GAD7 (*g* = −0.54) and small for PHQ8 (*g* = −0.48).

### 3.4. Preliminary Efficacy on the Primary Outcome

There was a significant interaction between the trial arm and time (*p* < 0.05). Follow-up analyses showed wellbeing significantly improved over time in the intervention arm (F(2, 2042.221) = 12.75, *p* < 0.0001, partial η^2^ = 0.34) but not in the waitlist arm F(2, 3506.22) = 0.83, *p* = 0.44, partial η^2^ = 0.04). See Figure 2 for an illustration of the change over time by trial arm.

### 3.5. Reliable Change and Clinically Significant Change

The reliable change results for the completers are presented in Figure 3. One participant made zero change on the primary outcome, and a further two made no reliable change. The remainder (14) made reliable improvements in wellbeing (82%), and 10 (59%) met the criterion for CSC. None deteriorated (see Figure 3).

## 4. Discussion

The results of this pilot found *Roots and Shoots* to be reasonably feasible and well accepted by participants. Preliminary evidence suggests *Roots and Shoots* may be effective in increasing mental wellbeing in this population. A large effect size (*g* = 1.24) for mental wellbeing was found in favour of the intervention. The majority made reliable improvements in wellbeing (82%), and 10 (59%) met the criterion for CSC. None of the intervention participants deteriorated. This finding is consistent with meta-analyses of nature-based activities ([20]). Effects for secondary outcomes tended to be smaller and generally favoured the intervention. Within these mostly medium-sized effects, the larger ones were seen in relative increases in mindfulness (*g* = 0.70), stress reduction (*g* = −0.67) and increased nature connection (*g* = 0.61). For stress and mindfulness, the confidence intervals did not overlap zero in this imputed data set. The effect for sleep disturbance was small (*g* = −0.17). Interestingly, average scores on the clinical measures (PHQ8 and GAD7) suggested that the intervention group moved from clinical caseness for both depression and anxiety to “not caseness” at T2. The effect size for change over time was medium for GAD7 (*g* = −0.54) and small for PHQ8 (*g* = −0.48).

The effects found in this pilot were not dissimilar to those found in previous full trial research. For example, a review of studies offering cognitive–behavioural self-help (self-administered, minimal contact or guided) reported an overall Hedges’ *g* on the primary outcome of *g* = −0.49, with little difference between self-administered (*g* = −0.42), minimal (*g* = −0.55) and guided (*g* = −0.53) ([29]). Depending on the results of the next *Roots and Shoots* RCT, a future non-inferiority trial could compare this intervention with a cognitive–behavioural approach.

Overall, participants reported high engagement and adherence to the intervention, with 88.2% of participants completing the guidebook for 50% or more days of the four-week intervention period, which is congruous with other self-help interventions for wellbeing ([50]). These findings are promising considering self-help interventions rely heavily on an individual’s willingness to engage with the intervention, which is often associated with lower adherence than guided interventions ([33]). It should be noted that the ITT approach used here provides unbiased estimates of the efficacy of the intervention on the primary outcome but will underestimate the magnitude of the efficacy for adherent participants ([55]). In a full trial, consistent with other mental health intervention RCTs ([5]), it will be helpful to additionally run Complier Average Causal Effect (CACE) analyses. CACE assesses the causal effect of an intervention for those receiving it as intended, adjusting for rates of intervention adherence.

The present study is consistent with previous research that supports both the effectiveness of self-administered interventions for mental health difficulties, such as depression ([87]), and their comparability with guided interventions ([42]). One advantage of self-help interventions is that they can reduce the perceived social stigma that is often associated with other approaches to mental health ([72]). Nevertheless, it is important to acknowledge that there may be social and motivational advantages to guided approaches, particularly in individuals with poor mental health or wellbeing ([77]).

The data in the present study suggest that the general consensus from participants was that the guidebook was helpful and easy to use, with nearly all participants (94%) specifying they would continue using the intervention and recommend *Roots and Shoots* to a friend. Interestingly, in terms of intervention duration, although some participants expressed a preference for a shorter duration, a sizeable proportion of participants (47%) suggested the intervention should be longer than 4 weeks. Consideration of this is warranted given that the duration of cognitive–behavioural therapy (CBT)-informed self-help interventions, with minimal contact support, typically ranges between four and 11 weeks, with the large majority of interventions lasting 4 to 7 weeks ([29]). Future studies should test the optimal duration for the intervention. For example, a 2-week version could be tested, or a study could be designed to compare the efficacy of the 4-week version with a longer intervention protocol (e.g., 8 weeks). Even a longer duration may be beneficial, given that a 10-week nature-based therapy for those with stress-related illness resulted in improved wellbeing at a 12-month follow-up ([84]).

While it was beyond the scope of the present study to carry out a full qualitative analysis, preliminary feedback was positive. However, future work in developing the intervention should adopt a mixed-methods approach to assess the acceptability and feasibility of the intervention in both quantitative and qualitative terms. For example, a single-arm, mixed-methods study could use focus groups to explore participant experience in depth.

The target sample size was achieved in the present pilot study, although there were a number of challenges in recruiting and retaining participants, which are well-documented issues associated with trials ([94]). First, despite the inclusion of CAPTCHA, inspection of screener responses suggested a number of suspected automated bot responses ([70]; [100]), which may account for low uptake following completion of the screener. It is worth noting that, in the present study, an attempt to screen out automated bot responses was also made via the requirement of a valid postal address and email confirmation of this prior to commencement. Email review has been suggested as one of the most effective strategies for resolving bot issues ([100]). Nonetheless, whilst subsequent trials should employ similar recruitment approaches via social media platforms, a viable strategy to reduce bots may include direct survey links in advertisements to be replaced with requests for prospective participants to contact researchers directly for survey links. The attrition rate was substantial overall in this pilot study (36%), most notably in the waitlist arm (43%) when compared with the intervention arm (29%). The rate of attrition, although greater than we initially expected, is, however, broadly consistent with previous research on self-guided, remote, web-based interventions for mental health. For example, Karyotaki and colleagues found attrition rates were high for those completing less than 25% of modules (40%) and for those completing less than 50% of modules (59%) ([43]). Similarly, challenges with attrition in waitlist controls are commonly reported. For example, inactive control arms are associated with significantly higher rates of attrition compared to psychological placebos ([19]). Attrition in the present study’s waitlist arm is comparable with other studies investigating unguided self-help interventions. For example, a 4-week ACT-based guidebook reported 41% attrition in the waitlist control ([69]).

As commonly reported in psychological research, this sample included a significant majority of female participants. One possible explanation for this observation is that there are gender differences in attitudes towards help-seeking and willingness to participate in psychological treatments ([49]). Prior studies have noted an increased reluctance in help-seeking for mental health issues in males ([96]) and indicate associations between socially constructed beliefs of masculinity, including stoicism, self-reliance and restrictiveness, and a lower likelihood of uptake and engagement with mental health treatment ([76]). Thus, emphasis on gender-sensitive interventions that utilise masculine norms, male-specific role models ([74]) and solution-focused approaches in advertising mental health interventions ([51]) have been shown to improve help-seeking attitudes. As an important issue within psychological research, future investigations in this area may be tailored to increase male participation in mental health interventions.

Although not possible in the present study, future work should test for the mechanisms of action or “key ingredients” involved in producing effects in the intervention ([21]). The results in the present study must be interpreted with caution, but there is a suggestion that the intervention can lead to increases in positive relationships with nature as well as increases in mindfulness. This would be consistent with the nature theory and existing interventions, such as forest bathing, which has significant overlaps with mindfulness and is inherently meditative. Future studies could test for these potential mechanisms of action using mediation analysis. Guidelines also suggest that good theoretical understanding is essential to help with testing mechanisms of action ([21]). For example, recent theory suggests that a relational human–nature connection and attachment may explain the benefits of nature that may come from early experiences but equally from NBIs ([12]).

While we assessed the duration and the quality of engagement with self-guided intervention in the present study, previous research has adopted more comprehensive approaches. For example, in a process evaluation of a nature prescription study, social support, motivation, physical activity and nature use were tested to further understand the effects of the intervention ([66]). Such approaches offer more nuanced insights into the quality of participants’ interaction with the intervention beyond simple activity preference or completion.

Finally, it will be important to understand which elements of the intervention are driving the effects on mental wellbeing and for whom. This work may lead to a precision or tailored approach to intervention, which may in turn lead to larger effect sizes. For example, it may be that increasing good habits (e.g., spending more time in nature) may be more effective for some individuals, whereas developing more mindfulness in nature may be more effective for others. More broadly, the effectiveness and success of NBIs varies between individuals and can be influenced by a number of factors ([15]). For example, evidence suggests the efficacy of greenspace exposure may be contingent upon an individual’s personality, with the psychological benefits of nature exposure reduced in individuals who report high neuroticism and low conscientiousness ([1]). Furthermore, social factors, including accessibility to public natural spaces, can present a potential barrier to accessing and engaging with NBIs ([79]). [18] ([18]) have suggested that distance between an individual and their access to greenspace is an integral aspect affecting usage. In addition, disadvantaged communities have been shown to have significantly fewer natural areas of vegetation and parks, highlighting the significant role of social equity issues in one’s ability to engage with activities incorporated in NBIs ([79]). These issues should be addressed in future research.

### Clinical Implications

Economic benefits of NBIs from reduced public service use are estimated to be up to GBP 31,520 per person in one year ([71]), suggesting that this form of intervention, alongside others, could make a meaningful contribution to help reduce the burden on healthcare systems. This presents a reasonable solution, given mental health services continue to face increasing demand ([28]), especially post the COVID-19 pandemic ([80]). The integration of nature-based techniques into clinical practice has promising potential, particularly in addressing mental health concerns like anxiety, stress, rumination and depression ([7]; [63]) and improving wellbeing ([16]). Following further studies and robust conclusions, clinicians could consider incorporating NBIs, such as the present intervention, into their practice to directly address symptoms of stress, anxiety and rumination. The intervention could be used to enhance and support mindfulness practice, which in itself has utility in improving a range of psychological and health conditions ([102]). The integrated BA principles and symptom and habit monitoring present a potentially helpful and pragmatic option for home practice outside of therapeutic sessions. The intervention may have applications as a standalone intervention or as part of an integrative approach. Health services may also consider using this intervention as part of a broader “stepped care” model ([53]). This could potentially be used as both step 1 (watchful waiting, pure self-help and early intervention) and step 2 (guided self-help). It could also have utility for people being “stepped down” from more intensive therapy (e.g., from step 3: high-intensity therapy) or even as part of a public health strategy, incorporating ongoing preventative or early interventive lifestyle change. It should be noted that while recruitment was based on relatively low levels of mental wellbeing, the present sample was not drawn from a clinical population, per se. However, average PHQ scores reported in this study were consistent with “probable depression” (i.e., ≥10). In the UK, National Health Service (NHS) Talking Therapies use PHQ scores ≥10 to determine clinical “caseness”. Nevertheless, it will be important to adapt and test the intervention with clinical populations (e.g., those referred to mental health service support with major depressive disorder (MDD)).

Brief self-help NBIs, like *Roots and Shoots*, offer a low-risk and cost-effective option and can be applied to diverse settings, from urban environments to more immersive nature experiences or even virtual exposure. This flexibility can support those with unequal access ([34]) or for whom it may be prohibitive to access nature, for example, those in hospital beds or prisons. The flexibility of content inherent in the intervention allows for personalised care plans, potentially improving patient engagement and adherence to therapeutic interventions.

## 5. Conclusions

The main goal of the current study was to determine the feasibility and acceptability of a novel, self-help NBI to improve mental wellbeing. Overall, the results suggest that a future fully powered RCT should be feasible, and the present data on efficacy suggest that this is warranted.

## Figures and Tables

**Figure 1 behavsci-15-01096-f001:**
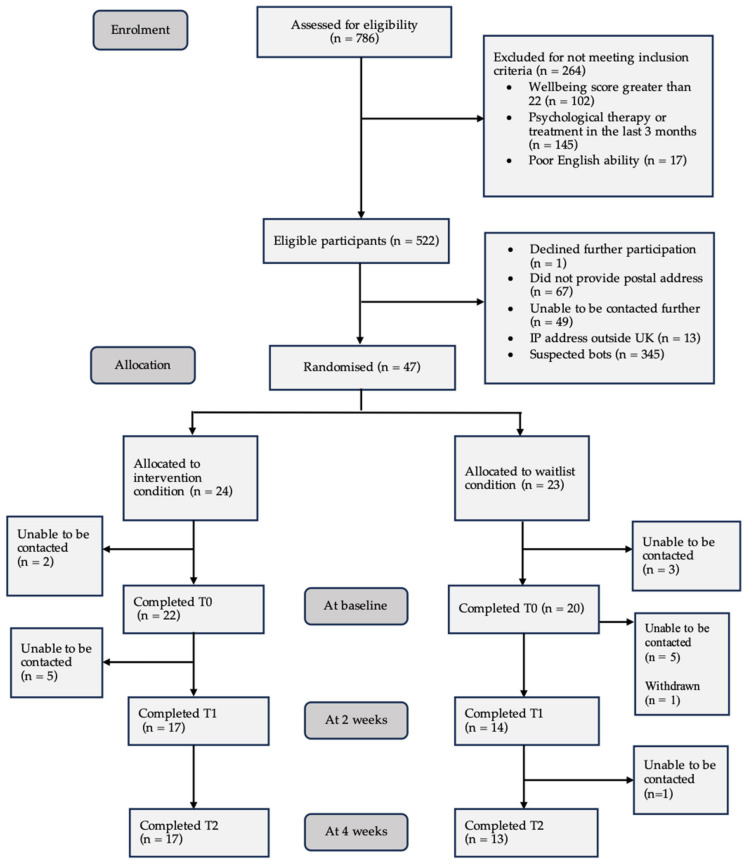
CONSORT diagram showing the flow of participants through the study.

**Figure 2 behavsci-15-01096-f002:**
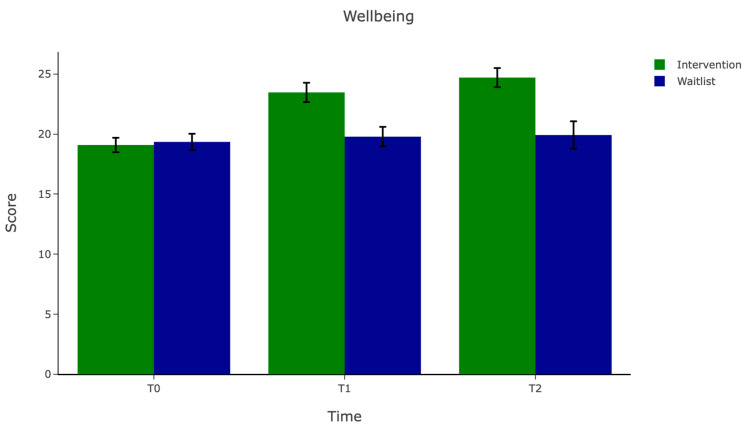
An illustration of the change in mental wellbeing by trial arm. Error bars represent 1 ± standard error of the mean.

**Figure 3 behavsci-15-01096-f003:**
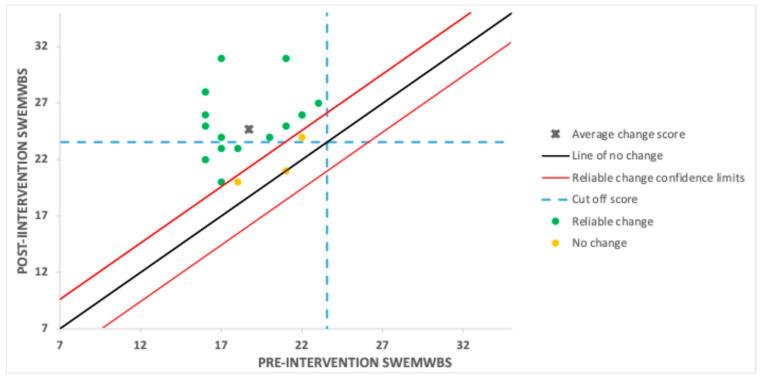
Reliable and clinically significant change in the intervention arm.

**Table 1 behavsci-15-01096-t001:** Demographic characteristics at baseline by trial arm.

VARIABLE	Intervention Group (*n* = 22)	Waitlist Group (*n* = 20)	*p* Value
**Age, *n* (%)**			0.41
** 18–24**	3 (13.64)	5 (25.00)	
** 25–34**	4 (18.18)	2 (10.00)	
** 35–44**	6 (27.27)	8 (40.00)	
** 45–54**	8 (36.36)	3 (15.00)	
** 55–64**	1 (4.54)	2 (10.00)	
**Gender, *n* (%)**			0.16
** Female**	21 (95.50)	15 (75.00)	
** Male**	1 (4.5)	4 (20.00)	
** Non-binary/third gender**	0 (0.00)	1 (5.00)	
**Ethnicity, *n* (%)**			0.37
** White**	21 (95.45)	19 (95.00)	
** Asian**	1 (4.55)	0 (0.00)	
** Prefer not to say**	0 (0.00)	1 (5.00)	
**Ses, *n* (%)**			0.30
** High**	0 (0.00)	2 (10.00)	
** Middle**	14 (63.64)	12 (60.00)	
** Low**	5 (22.73)	2 (10.00)	
** Prefer not to say**	1 (4.55)	2 (10.00)	
** No response**	2 (9.10)	2 (10.00)	0.80

**Table 2 behavsci-15-01096-t002:** Participant responses to acceptability and feasibility items.

	Strongly Disagree n (%)	Disagree n (%)	Neutral n (%)	Agree n (%)	Strongly Agree n (%)
**Overall, I enjoyed doing the Roots and Shoots programme**					
	0 (0.00)	0 (0.00)	0 (0.00)	10 (58.82)	7 (41.18)
**I would come back and use the Roots and Shoots guidebook again in the future**					
	0 (0.00)	0 (0.00)	1 (5.88)	8 (47.06)	8 (47.06)
**The Roots and Shoots programme was helpful in improving my wellbeing**					
	0 (0.00)	0 (0.00)	2 (11.76)	10 (58.82)	5 (29.41)
**I would recommend the Roots and Shoots programme to a friend**					
	1 (5.88)	0 (0.00)	0 (0.00)	9 (52.94)	7 (41.18)
**The information in the guidebook was helpful**					
	0 (0.00)	0 (0.00)	1 (5.88)	11 (64.71)	5 (29.41)
**The habit wheel was helpful**					
	0 (0.00)	0 (0.00)	4 (23.53)	8 (47.06)	5 (29.41)
**The habit and mood tracker was helpful**					
	1 (5.88)	0 (0.00)	2 (11.76)	7 (41.18)	7 (41.18)
**The expressive writing tool was helpful**					
	0 (0.00)	3 (17.65)	3 (17.65)	7 (41.18)	4 (23.53)
**The coping tool was helpful**					
	1 (5.88)	0 (0.00)	4 (23.53)	10 (58.82)	2 (11.76)
**The honeycomb challenge was helpful**					
	0 (0.00)	(0.00)	8 (47.06)	6 (35.29)	3 (17.65)
**I found the nature videos helpful**					
	0 (0.00)	0 (0.00)	5 (29.41)	9 (52.94)	3 (17.65)
**I found the meditations helpful**					
	0 (0.00)	1 (5.88)	6 (35.29)	8 (47.06)	2 (11.76)
**I found the breathing exercises helpful**					
	0 (0.00)	0 (0.00)	3 (17.65)	8 (47.06)	6 (35.29)

**Table 3 behavsci-15-01096-t003:** Means, 95% confidence intervals and effect sizes.

Measure	T0	T0 Difference by Arm	T1	Change from Baseline (T0–T1)	T1 Effect Size Between Group	T2	Change from Baseline (T0–T2)	T2 Effect Size Between Group
	**Mean (sd)**	**Hedges’ g (CI)**	**Mean (sd)**		**Hedges’ g (CI)**	**Mean (sd)**		**Hedges’ g (CI)**
**SWEMWBS**								
Intervention	19.09 (2.83)	−0.09 (−0.68 to 0.51)	23.25 (3.16)	4.16	1.08 (0.42 to 1.73)	24.36 (3.02)	5.27	1.24 (0.57 to 1.91)
Waitlist	19.35 (3.03)		20.30 (3.93)	0.95		20.92 (4.08)	1.52	
**GAD**								
Intervention	9.40 (4.63)	0.05 (−0.55 to 0.64)	6.93 (3.92)	−2.47	−0.45 (−1.07 to 0.17)	5.46 (3.31)	−3.94	−0.54 (−1.16 to 0.08)
Waitlist	9.20 (3.91)		8.71 (4.73)	−0.49		7.63 (3.27)	−1.57	
**PHQ**								
Intervention	10.39 (4.64)	0.02 (−0.58 to 0.61)	8.42 (5.03)	−1.97	−0.23 (−0.84 to 0.38)	7.25 (4.38)	−3.14	−0.48 (−1.10 to 0.14)
Waitlist	10.30 (4.40)		9.38 (4.48)	−0.92		9.38 (4.61)	−0.92	
**PSS**								
Intervention	9.09 (1.93)	0.17 (−0.43 to 0.76)	7.36 (1.84)	−1.73	−0.78 (−1.41 to −0.14)	6.36 (1.77)	−2.73	−0.67 (−1.30 to −0.04)
Waitlist	8.73 (2.30)		8.67 (1.73)	−0.06		7.45 (2.28)	−1.28	
**RRS**								
Intervention	12.07 (2.95)	−0.22 (−0.82 to 0.37)	11.52 (3.12)	−0.55	−0.07 (−0.68 to 0.54)	9.72 (3.31)	−2.35	−0.56 (−1.19 to 0.06)
Waitlist	12.76 (3.14)		12.43 (3.28)	−0.33		12.16 (2.93)	−0.60	
**JSS**								
Intervention	11.75 (4.88)	−0.14 (−0.73 to 0.46)	11.95 (4.23)	0.20	0.24 (−0.37 to 0.85)	9.42 (4.68)	−2.33	−0.17 (−0.78 to 0.45)
Waitlist	12.44 (5.08)		11.43 (5.39)	−1.01		10.95 (5.60)	−1.49	
**FFMQ**								
Intervention	34.79 (5.76)	0.19 (−0.41 to 0.78)	35.32 (4.73)	0.53	0.12 (−0.49 to 0.73)	39.98 (5.50)	5.19	0.70 (0.07 to 1.33)
Waitlist	33.56 (7.02)		33.33 (6.70)	−0.23		34.16 (7.00)	0.60	
**NR6**								
Intervention	3.95 (0.78)	−0.30 (−0.90 to 0.29)	4.11 (0.63)	0.16	0.26 (−0.35 to 0.88)	4.46 (0.50)	0.51	0.61 (−0.02 to 1.23)
Waitlist	4.18 (0.70)		4.14 (0.89)	−0.04		4.23 (0.70)	0.05	

## Data Availability

The data presented in this study are available on request from the corresponding author.

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
