# Peer review of "Roots and Shoots: A Pilot Parallel Randomised Controlled Trial Assessing the Feasibility and Acceptability of a Nature-Based Self-Help Intervention for Low Wellbeing"

_behavsci, 2025, doi:10.3390/bs15081096_

Round 1

Reviewer 1 Report

Comments and Suggestions for Authors

The MS presents a very interesting and well-developed study to assess acceptability and feasibility of a self-administered nature-based intervention.  The topic is actual and with many implications. Thus I am positive towards publication pending the minor revisions listed below

Procedure. Did you collect measures on duration, quality, other details of the self-guided activities done? If not please add in the Limitations and future Avenues sections

Discussion. Your rationale seems to suggest that self-administered lead to similar effects than guided. I would report at least about the social and motivational advantages to run guided activities.

You also report about a once a week activity effectiveness. What about more enduring programs? For instance

https://journals.sagepub.com/doi/10.1177/03946320070200S202

https://www.mdpi.com/1999-4907/14/7/1423

As usual, women mostly participated. I would add in the Discussion some insights on how favoring males participation.

Individual differences in effectiveness are clearly documented. Any insight on the factors leading to them? The place where the activities were run? Psychological factors? Individual preferences? Others?

I wish the AA the best with their research!

Reviewer 2 Report

Comments and Suggestions for Authors

This is a very clear and well written article. The context and rationale are well developed  and evidenced with valid citations. 

The methodology is  clear, however the consort diagram seems to have sections missing or could be presented more clearly for  the reader  Line 109 - Formal power calculations are not recommended for pilot studies (Whitehead et al., 2016), given that one of their functions is to gain initial estimates for sample size in future 
trials (Lancaster, 2015). There was 

It would be valuable  to see the qualitative data analysed further, maybe in another article ?

Discussion of data and conclusions a re measured but highlight he implications of the Root and Shoots intervention and  as suggested  point to the benefits of conducting a RCT trail.
